# TRAUMA THOMPSON: A NOVEL DATASET AND BENCHMARKS FOR AI COPILOTS FOR HUMANITARIAN OPERATIONAL MEDICINE

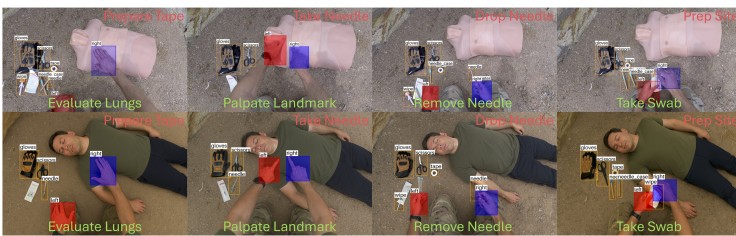

Figure 1: Action recognition (green), action anticipation (red), object detection, and hand tracking on Trauma THOMPSON dataset.

## ABSTRACT

This paper introduces the Trauma THOMPSON dataset, a novel dataset and benchmarks to foster research towards designing artificial intelligence-based decision-making algorithms specifically suited for life-saving interventions performed by less experienced caregivers. This paradigm is particularly relevant to support humanitarian operational medicine, where the essential resources are either unavailable or significantly restricted. Our dataset includes a total of 3717 high-resolution clips and ground-truth action annotations by medical professionals. The events are unscripted, and the clips are all assigned a specific medical skill relevant to life-saving interventions. There are two skill types: regular procedures with standard medical tools and improvised procedures with daily objects. We have augmented this dataset with additional annotations, including medical visual question answering, hand tracking and object detection. Moreover, we propose a framework for replacing manikins in the dataset with real patients and a realism detection method. Benchmarks are provided for action recognition, action anticipation, and visual question answering (VQA) using a variety of vision models and vision language models (VLMs). We found that MViT v2 is the best performer for action recognition and action anticipation and BLIP for VQA. By consolidating diverse annotations into a single dataset and a framework to create realistic patient images, Trauma THOMPSON dataset offers a foundation for training unified VLMs as AI medics that can perform holistic reasoning and decision-making in disconnected and high-stakes settings to support less experienced first responders. The dataset and codes are available at `https://dataverse.harvard.edu/previewurl.xhtml?token=bd66015d-64bc-4203-ad09-5ab5c90832ef`.

## 1 INTRODUCTION

In medicine, Artificial Intelligence (AI) assistants have been proposed to act as copilots for a mentee or trainee. Bahl et al., for example, have developed an AI system to guide radiologists (Bahl, 2020), and there is a wide body of work about the use of AI for diagnostics based on medical records and multimodal imaging techniques (Liu et al., 2018; Al-Antari, 2023; Dilsizian & Siegel, 2014; Hamet & Tremblay, 2017), which are in general offline approaches. The use of AI for mentoring

and intraoperative instruction is more recent (Dinh et al., 2023). For example, the Virtual Operative Assistant (Mirchi et al., 2020) offers feedback to trainees in neurosurgery during the resection of brain tumors in a VR setting. Likewise, Auloge et al. and Jha et al. (Auloge et al., 2020; Jha & MB, 2019) proposed assistance for spatial navigation during surgery. Many others have relied on AI as "evaluators" to assess surgical performance (Bissonnette et al., 2019; Ward et al., 2021; Fazlollahi et al., 2022). All such systems were developed for controlled environments, such as the Operating Room (Novaes & Basu, 2020), or alternatively were used in laboratory conditions (Rojas et al., 2020; Rojas-Muñoz et al., 2020; Zhang et al., 2021; Xu et al., 2022; Vannaprathip et al., 2025; Caballero et al., 2025). Recently, a first prototype for video analysis software based on speech recognition was created for operational medicine for first responders (Kar et al., 2021). However, the focus of that work is on data collection with Natural Language Processing (NLP).

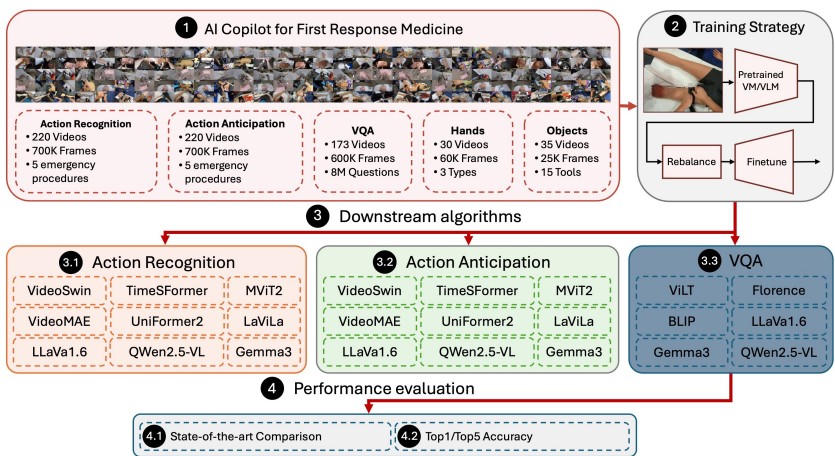

Figure 2: Overview of the experimental pipeline.

None of the aforementioned systems were used for humanitarian medicine, and neither did they address the specific challenges of uncontrolled field conditions. Some of those challenges involved just-in-time decision making with limited tools. To address the gap, we introduce the Trauma THOMPSON dataset and benchmarks, which are a collection of video clips with annotations and algorithms to encourage research and development of artificial intelligence (AI) copilots for humanitarian medicine. To the best of our knowledge, our dataset is the first of its kind in terms of scale, settings, challenges, and applicability. Figure 1 illustrates the action recognition, action anticipation, object detection, and hand tracking tasks on the Trauma THOMPSON dataset. The first row shows the tasks performed on manikin images and the second row shows the tasks performed on realistic images generated through an image generation framework we propose. Figure 2 shows an overview of the experimental pipeline of this study. In summary, this paper makes the following contributions.

- We created the first egocentric view dataset of operational medicine that assist field medics to properly perform emergency care procedures in resource-constrained settings. This dataset includes annotated video clips corresponding to 5 unscripted life-saving procedures.

- We provide benchmarks for action recognition and anticipation to predict the actions required for humanitarian medicine and resuscitative care with multiple vision models (VMs) and vision language models (VLMs). This dataset is intended to act as an essential piece for developing copilots for medics and first responders.

- We created secondary annotations for VQA and provide benchmarks for this task to illustrate how VQA algorithms can be potentially used as a clinical decision support (CDS) tool to assist caregivers through natural dialogue throughout the diagnostic process.

- We tested a methodology to generate pseudo-realistic images of patients in a variety of settings from images using simulators, and realism assessment metrics. It addresses the challenge of real patient data collection under emergency conditions without identity concerns.

## 2 RELATED WORK

### 2.1 EGOCENTRIC ACTIVITY RECOGNITION DATASETS

Video understanding has seen dramatic advances due to the introduction of action classification benchmarks such as UCF101 (Soomro et al., 2012), HMDB51 (Kuehne et al., 2011), Kinetics (Kay et al., 2017), Something-Something (Goyal et al., 2017), and AVA (Gu et al., 2017), which mostly consist of short videos focusing on a single action per clip and aim to recognize daily activities. Nonetheless, these datasets may lack the spontaneity, progression, and multi-tasking that occur in real-life situations due to their scripted nature. As a result, research has shifted focus to first-person vision, which delivers activities from a unique viewpoint. For instance, Pirsiavash & Ramanan (2012) developed a dataset that includes 20 participants and encompasses 10 hours of ADL videos. Li et al. (2020) created EGTEA Gaze+ with wearable cameras, which is an egocentric ADL dataset of 28 hours of cooking activities from 86 distinct sessions involving 32 subjects. Damen et al. (2018) curated the EPIC-KITCHENS dataset, which is a large-scale egocentric video dataset with 100 hours of cooking actions recorded by 32 participants. These egocentric-view datasets often are used for open challenges in action recognition and action anticipation, as is the case with the EPIC-KITCHENS dataset.

### 2.2 SURGICAL AND FIRST RESPONDERS DATASETS

With the surge of machine learning techniques, there has been a renewed interest in the acquisition of surgical datasets, as they provide fundamental resources for post-recognition error, accuracy, and further procedural correction. Furthermore, data collection has enabled the computation of proficiency, skill level, knowledge acquisition, and performance during robotic surgery (Gao et al., 2014; Tao et al., 2012; Gonzalez et al., 2021). Moreover, with the advance in natural language processing, AI assistants have been proposed to assess, assist, and coach nurses, residents, and assistants using behavior understanding (Lee & Yoon, 2021) and captioning methods to communicate next steps (Hartmann et al., 2022). Instructional videos for life-saving skills have been proposed for the purpose of training AI algorithms (Gupta et al., 2023). Nevertheless, such datasets were collected in controlled settings using both simulation and planned surgical procedures. There are very few cases in that such datasets were collected under austere and limited conditions, as often found in humanitarian and operational medicine (Wang et al., 2021).

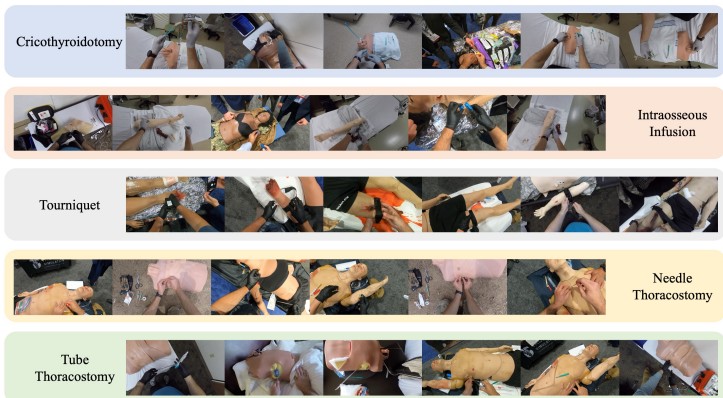

Figure 3: Examples of procedure video clips.

### 2.3 ACTION RECOGNITION AND ACTION ANTICIPATION

Many researchers have represented and recognized human actions through various visual (including RGB video, skeleton, depth, infrared, point cloud, event stream) and non-visual (including audio, acceleration, radar, and WiFi) modalities. Even with challenges from varying viewpoints, scales, backgrounds, and illuminations, RGB video remains the most popular, accessible, and single-modality method to capture and represent actions (Sun et al., 2023). For the action recognition task, with

Table 1: Comparison of Trauma THOMPSON to the related egocentric and medical datasets

| Dataset | Ego | Med | Frames | No. Act | Participants | No. Envs |
|---|---|---|---|---|---|---|
| **Trauma THOMPSON, 2025** | ✓ | ✓ | 0.7M | 162 | 12 | 15 |
| EPIC-KITCHENS (Damen et al., 2018) | ✓ | ✗ | 11.5M | 149 | 32 | 32 |
| BEOID (Damen, 2014) | ✓ | ✗ | 0.1M | 34 | 5 | 1 |
| GTEA (Fathi et al., 2011) | ✓ | ✗ | 0.4M | 42 | 13 | 1 |
| CMU-MMAC (de la Torre et al., 2008) | ✓ | ✗ | 0.2M | 31 | 16 | 1 |
| ADL (Pirsiavash & Ramanan, 2012) | ✓ | ✗ | 1.0M | 32 | 20 | 20 |
| ESAD (Bawa et al., 2020) | ✗ | ✓ | 0.03M | 21 | 4 | 4 |
| CholecT50 (Nwoye et al., 2022) | ✗ | ✓ | 0.1M | 100 | 13 | 13 |
| MedVidCL (Gupta et al., 2023) | ✗ | ✓ | 1489 Videos | 0 | >100 | >100 |
| MRAO (Schmidt et al., 2021) | ✗ | ✓ | 480 Videos | 10 | 16 | 2 |
| MISAW (Huaulmé et al., 2021) | ✗ | ✓ | 27 Videos | 17 | 6 | 1 |
| PSI-AVA (Valderrama et al., 2022) | ✗ | ✓ | 8 Videos | 167 | 3 | 1 |
| PETRAW (Huaulmé et al., 2023) | ✗ | ✓ | 150 Videos | 6 | 4 | 2 |

the wide adoption of deep learning, a range of architectures has been proposed in the literature, mainly based on convolutional neural networks (CNN), recurrent neural networks (RNN), and vision transformers (ViT) (Yang et al., 2022; Ulhaq et al., 2022). CNN methods are classified into 2D CNN and 3D CNN, while RNN methods typically use gated architectures like Long-Short Term Memory (LSTM). ViT has become prominent in video understanding due to its attention mechanism. Predicting future actions is key for AI copilots. Thus, the action anticipation task can be divided into three categories: early action recognition; next action anticipation; and long-term action anticipation (Roy et al., 2023). Various methods (Kong & Fu, 2022) including Large Language Models (LLMs) and Large Vision Language Models (LVLMs) have been explored to tackle these tasks (Yan et al., 2024; Xiang et al., 2023; Dessalene et al., 2023; Wang et al., 2025).

### 2.4 MEDICAL VQA DATASETS

Recent advancements in medical VQA include the VQA-MED-2018 (Hasan et al.), VQA-MED-2019 (Abacha et al., 2019), VQA-MED-2020 (Abacha et al., 2020), VQA-MED-2021 (Ben Abacha et al., 2021), PathVQA (He et al., 2020), VQA-RAD (Lau et al., 2018), RadVisDial (Kovaleva et al., 2020), and SLAKE (Liu et al., 2021). These datasets typically focus on medical imagery such as CT scans, MRI, x-ray, and ultrasound. VQA can serve as a valuable CDS tool by enabling a sequential approach to diagnostic processes, coming up with conclusions, and potentially facilitating diagnostic inferences (Zhang et al., 2024). Foundational Models with Chain-of-Thought architectures have been recently proposed to mimic the step by step reasoning process (Kim et al., 2024; Xu et al., 2024). However, a key challenge in this area lies in the scarcity of adequate and diverse training data, which is essential to create robust AI algorithms under real-world scenarios (Antoniadi et al., 2021). VQA datasets for resource-limited settings are even more scarce, yet they hold significant potential for enhancing life-saving capabilities.

### 2.5 OUR WORK: EGOCENTRIC OPERATIONAL MEDICINE DATASET

Table 1 compares the Trauma THOMPSON dataset to common egocentric view and medical instructional datasets and presents key metrics that distinguish the Trauma THOMPSON dataset as the first egocentric view medical instructional dataset with per-frame annotations. The dataset has a similar structure as other egocentric datasets for action recognition and anticipation, such as EPIC-KITCHENS (Damen et al., 2018), GTEA (Fathi et al., 2011) and EGTEA Gaze+ (Li et al., 2020), Charades-Ego (Sigurdsson et al., 2018) dataset, with the following caveats. Firstly, the hands are not always visible or distinguishable due to artificial blood, occlusions, and multiple limbs, which makes it more challenging for detection and tracking. Secondly, some of the videos are taken outdoors, increasing the complexity due to uncontrolled lighting. Thirdly, mistakes require rewinding and re-doing, or stopping short while completing the procedures, leading to a high variability in style and performance time. Additionally, as opposed to existing datasets for surgical guidance and instruction, which rely on a fixed set of tools common to general surgery, our dataset is subject to the emergency setting, as shown in Figure 3. That is, performers were first responders, medics, and surgeons; and the procedures were often conducted using improvised tools (e.g. shirt for a tourniquet, a pocket knife for a cricothyroidotomy) to replicate a resource-limited setting. Lastly, our dataset includes pseudo-realistic images of patients generated using a VLM.

## 3 METHOD OF DATA COLLECTION

**Procedure identification** The development of the Trauma THOMPSON dataset involved a team of experts with experience in deployed settings, such as surgeons, critical care physicians, and emergency medicine physicians, who created a list of essential procedures for prolonged casualty care (PCC), such as cricothyrotomy and tourniquet application. Additionally, a focus group of 15-30 subject matter experts (SMEs) determined a consensus on the content and best practices for the dataset. This information was used to identify a final list of procedures, which includes cricothyroidotomy, intraosseous infusion, tourniquet, needle thoracostomy, and tube thoracostomy. The collection of procedures and settings is described in the following section.

**Data collection** We focused on capturing natural, unscripted life-saving intervention (LSI) procedures from the first-person perspective, which involves operating a medical tool, searching for an item, changing one's mind, and encountering unexpected problems. The videos were recorded at 1080p using head-mounted cameras (GoPro, Hero7, San Mateo, California) to capture first-person views filmed across various simulation models and environments. Surgeons wore the cameras on their heads and adjusted the angle to 20-30° relative to the forehead for optimal video collection. The hands were centered in frames during procedures for better visualization.

The dataset was also enhanced through the inclusion of videos capturing "just-in-time" (JIT) procedures involving improvised, non-traditional equipment. Videos were obtained of users performing improvised tourniquets (utilizing belts or clothing and a screwdriver), tube thoracostomy (utilizing scissors for incision and expansion of thoracostomy and a screwdriver to guide insertion of the tube), needle cricothyroidotomy (replacing standard incision/tube with a needle for emergency airway management), and manual intraosseous needle placement (when the needle driver is not available or functional). In addition to the egocentric manikin recordings, we curated a complementary set of real human emergency procedure videos sourced from publicly available YouTube content. These videos depict the five life-saving interventions in the Trauma THOMPSON dataset on real human subjects. Each video was manually reviewed to ensure clear visualization of the procedure types. To further expand the dataset and improve translation to clinical scenarios, we generated pseudo-realistic synthetic images using an AI-based image generation model. These images depict realistic trauma presentations, including active bleeding, soft tissue deformation, and blunt-force injuries, enabling more realistic cases. The details is in Section 4.

**Annotation pipeline** The action annotations in the dataset consist of start timestamps, end timestamps, and actions expressed as verb-noun pairs for corresponding video clips. The expected output for testing is the labels for the action, verb, and noun. Medical professionals were responsible for annotating the data, and providing the timestamps and actions for each procedural step. To reduce the possibility of errors in timestamping and video segmentation, the annotations underwent peer review. The medical VQA is derived from the egocentric video dataset and includes additional annotations that contain questions and corresponding plausible answers. Each question in the secondary annotations contain 3 to 5 potential answers. For example: *Q: What limb is injured? A: Right arm; Q: Where is the catheter insderted? A: There is no catheter; A: Is there any bleeding? A: No.*

**Data Quality Assurance** To ensure annotation accuracy, each procedure is labeled by one medical professional and reviewed by two others. Since exact timestamps are difficult to determine, we estimate the *actual* timestamp as the average of all annotators' inputs. Annotation accuracy is then calculated based on the overlap between the original and actual timestamps, relative to the actual duration of each clip. The average annotation accuracy across all clips is computed as a duration-weighted mean. Due to the large volume of the dataset, the annotation reviewers were requested to randomly review 60 videos in the dataset according to the above-stated instructions. The temporal accuracy is 99.4%, the label accuracies of actions, verbs, and nouns, are 97.2%, 97.2%, and 97.7%.

## 4 REALISM: REALISTIC IMAGE GENERATION

We introduce the **Trauma THOMPSON Framework (TTFW)**, a vision-language guided image-to-image generation framework designed to convert clinical images containing non-realistic elements—such as manikins or synthetic anatomical proxies—into photorealistic depictions suitable for both human interpretation and downstream machine learning tasks. The framework ensures that the

generated image remains semantically consistent with the source while satisfying perceptual **realism constraints**.

## 4.1 PROBLEM FORMULATION

Given a source image $I_S$, our objective is to generate a realistic image $I_R$ such that the clinical context is preserved, while any unrealistic subject (e.g., a manikin) is replaced with a human-like, photorealistic counterpart. We begin by computing a caption $C_S = \text{VLM}(I_S)$ using a vision-language model. To abstract away subject-specific tokens (e.g., "manikin"), we apply a masking function $\mathcal{M}$, yielding a subject-neutral prompt $\tilde{C}_S = \mathcal{M}(C_S)$. This prompt and the original image are then passed to a generative model $G$, which produces the output image $I_R = G(I_S, \tilde{C}_S)$. To ensure semantic consistency, we caption the generated image $C_R = \text{VLM}(I_R)$ and apply the same masking operation to obtain $\tilde{C}_R = \mathcal{M}(C_R)$. Using a language-image embedding function $\phi$, we define the contextual similarity as the cosine similarity between masked caption embeddings. The generated image is accepted only if $\cos(\phi(\tilde{C}_S), \phi(\tilde{C}_R)) \geq \delta$, where $\delta \in [0, 1]$ is a predefined similarity threshold (e.g., $\delta = 0.99$). If this constraint is not satisfied, we compute a semantic delta $\Delta_C = \text{LM}(C_S, C_R)$ using a language model, and iteratively refine $\tilde{C}_S$ before re-generating $I_R$. For our experiments, we used ChatGPT-4o for VLM, $G$, and LM.

**Realism Constraint.** We further require that the output image satisfies human-perceptual realism based on six binary criteria. Specifically, we define a realism function $\mathcal{R}(I_R) = [r_1, \ldots, r_N] \in \{0, 1\}^N$, where each component evaluates whether *the patient appears to be moving, whether there is a visible lesion or wound, the presence of bodily fluids, the naturalness of the skin texture, the completeness and anatomical coherence of the body, and whether the subject appears human.* The image is accepted only if $\sum_{i=1}^N r_i \geq \tau$. Algorithm 1 describes the process in detail.

## 4.2 ALGORITHM

---
**Algorithm 1** TTFW: Context-Guided Realistic Image Generation
---
**Require:** Source image $I_S$, VLM, generator $G$, masking function $\mathcal{M}$, realism metric $\mathcal{R}$, thresholds $\delta, \tau$
**Ensure:** Realistic output image $I_R$
1: $C_S \leftarrow \text{VLM}(I_S)$            ▷ Generate caption for input
2: $\tilde{C}_S \leftarrow \mathcal{M}(C_S)$            ▷ Mask subject-specific terms
3: **repeat**
4:      $I_R \leftarrow G(I_S, \tilde{C}_S)$            ▷ Generate image from prompt
5:      $C_R \leftarrow \text{VLM}(I_R), \tilde{C}_R \leftarrow \mathcal{M}(C_R)$
6:      $\text{sim} \leftarrow \cos(\phi(\tilde{C}_S), \phi(\tilde{C}_R))$            ▷ Compute similarity
7:      **if** $\text{sim} < \delta$ **then**
8:          $\Delta_C \leftarrow \text{LM}(C_S, C_R)$            ▷ Identify semantic drift
9:          $\tilde{C}_S \leftarrow \texttt{RefinePrompt}(\tilde{C}_S, \Delta_C)$
10:     **end if**
11: **until** $\text{sim} \geq \delta$
12: $r \leftarrow \mathcal{R}(I_R)$
13: **if** $\sum_{i=1}^N r_i \geq \tau$ **then**
14:      **return** $I_R$
15: **else**
16:      **return** `Failure:  realism criteria not met`
17: **end if**
---

## 4.3 REALISM METRIC AND EVALUATION

We define the realism score as the average of the N (=6 in our case) binary realism indicators: $\text{RealismScore}(I_R) = \frac{1}{N} \sum_{i=1}^N r_i$. This metric is used to validate generated images, either via expert raters or automated classifiers trained on trauma imagery datasets. Realism scores are reported both

as binary pass/fail and as continuous confidence levels, and can be correlated with downstream performance on medical AI tasks.

## 4.4 REALISM DETECTOR

By validating whether AI-generated images are indistinguishable from real patient imagery, realism detection confirms that augmented simulations maintain clinical relevance, bridging the gap between manikin practice and real-world scenarios. The failure of a classifier to differentiate between generated and real patient images demonstrates the high-fidelity images generated through TTFW. By gradually replacing more AI-generated frames into manikin videos, the realism classifier would fail. Firstly, a manikin video is decomposed into its individual frames. Then, we obtain a 6-dimensional realism feature vector from each frame with ResNet50 connected to a fully connected layer. Next, we average those frame-level embeddings to form a single video-level feature, which is fed into our pretrained Support Vector Machine (SVM) realism detector. If the video is still deemed a manikin, we invoke the TTFW pipeline to synthesize more realistic patient frames and inject them back into the sequence. This iterative process recomputes feature vectors, reclassifies with the SVM, and continuously injects generated frames until the detector finally labels the video as "real". Failure of the realism detector demonstrates the generated realistic images are very close to real human patients and can be useful for training AI models that require real human patients but not easily attainable.

## 5 RESULTS

### 5.1 BENCHMARK RESULTS FOR ACTION RECOGNITION AND ACTION ANTICIPATION

**Evaluation setups and metrics** The Trauma THOMPSON dataset was split into train and test sets with an 80/20 ratio for both the regular (177 videos) and JIT (43 videos) procedures. All algorithms were trained on either the regular or combined (regular + JIT) settings and evaluated under three categories: regular alone, JIT alone, and combined. The use of unconventional tools in the JIT procedures, coupled with the smaller size of the JIT set, negatively impacted algorithm performance. Each model was trained on a GeForce RTX™ 4090 Ti. For evaluation, a class-agnostic approach was applied to assess recognition and anticipation accuracy (Zhao et al., 2019), where a sequence of frames from the beginning of each action clip was sampled. Accuracy was measured as the ratio of correctly predicted clips to the total number of clips in the validation and test sets, with Top1 and Top5 metrics reported for verbs, nouns, and combined actions (verb + noun).

Table 2: Performance comparison (%) of action recognition models on different train-test settings.

| Train | Test | VideoSwin | | TimeSFormer | | VideoMAE | | Uniformer v2 | | MViT v2 | | LaViLa | |
|-------|------|------|------|------|------|------|------|------|------|------|------|------|------|
| | | Top1 | Top5 | Top1 | Top5 | Top1 | Top5 | Top1 | Top5 | Top1 | Top5 | Top1 | Top5 |
| Regular | Regular | 45.10 | 74.52 | 31.91 | 62.81 | 43.34 | 71.89 | 60.47 | 85.65 | **65.59** | **89.75** | 42.52 | 68.81 |
| | JIT | 3.85 | 15.38 | 0.51 | 5.77 | 5.77 | 17.31 | 8.65 | 21.15 | **14.49** | **35.20** | 0.96 | 9.62 |
| | Combined | 34.60 | 66.71 | 27.70 | 55.27 | 38.37 | 64.68 | 53.62 | 77.13 | **58.58** | **82.08** | 38.25 | 60.99 |
| Combined | Regular | 44.51 | 73.35 | 29.42 | 63.69 | 48.61 | 73.06 | 60.32 | 84.19 | **66.47** | **89.17** | 40.17 | 66.67 |
| | JIT | 39.42 | 70.19 | 32.69 | 58.65 | 44.23 | 65.38 | **53.85** | 80.77 | 50.96 | **90.38** | 37.71 | 63.84 |
| | Combined | 43.84 | 72.94 | 29.86 | 63.02 | 48.03 | 72.05 | 59.47 | 83.74 | **64.42** | **88.82** | 39.53 | 66.02 |

**Action recognition** All algorithms used were pretrained and then finetuned on the Trauma THOMPSON dataset. Table 2 presents a performance comparison of six action recognition models—VideoSwin (Liu et al., 2022), TimeSFormer (Bertasius et al., 2021), Uniformer v2 (Li et al., 2022a), VideoMAE (Tong et al., 2022), MViT v2 (Li et al., 2022b), and LaViLa (Zhao et al., 2022)—evaluated across three train-test scenarios: regular, JIT, and combined. Performance is reported using Top1 and Top5 accuracy, with per-procedure results provided in the appendix. In the regular train-test setting, MViT v2 achieves the highest accuracy (65.59% Top1, 89.75% Top5). When models are trained on the combined set, performance improves notably in the JIT test scenario, where Uniformer v2 achieves the best Top1 accuracy (53.85%) and MViT v2 reaches the highest Top5 accuracy (90.38%). The left image of Figure 4 shows the confusion matrix of the best-performing model, MViT v2, where the diagonal pattern indicates reliable predictions for frequent classes, while scattered dark spots in the lower portion highlight challenges in recognizing infrequent actions.

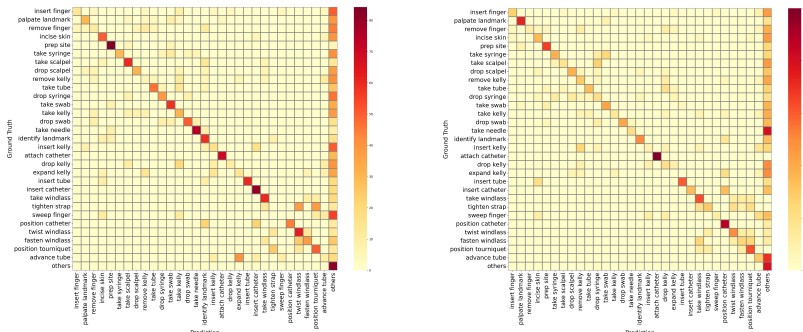

Figure 4: Comparison of confusion matrices for action recognition (left) and action anticipation (right) using MViT v2 on regular procedures.

**Action anticipation**    Table **??** presents the benchmarking results for action anticipation on the same six models. The performance on individual procedures is in the appendix. For the regular train-test setup, MViT v2 achieves the highest Top1 and Top5 accuracy of 60.12% and 87.02%. The action anticipation results follow a similar trend to action recognition, with MViT v2 and Uniformer v2 outperforming other models across all scenarios. However, the absolute performance metrics for anticipation are slightly lower than those for recognition. It suggests that predicting future actions is inherently more challenging than recognizing current actions, especially in diverse or unexpected scenarios such as JIT procedures. The right image of Figure 4 shows the confusion matrix of the best performing model MViT v2 on the action anticipation task. Similar performances are observed in action recognition, with less frequent classes being hard to classify for the model.

Table 3: Performance comparison (%) of action anticipation models on different train-test settings.

| Train | Test | VideoSwin | | TimeSFormer | | VideoMAE | | Uniformer v2 | | MViT v2 | | LaViLa | |
|---|---|---|---|---|---|---|---|---|---|---|---|---|---|
| | | Top1 | Top5 | Top1 | Top5 | Top1 | Top5 | Top1 | Top5 | Top1 | Top5 | Top1 | Top5 |
| Regular | Regular | 39.88 | 69.24 | 28.44 | 59.97 | 41.42 | 69.24 | 56.25 | 84.70 | **60.12** | **87.02** | 38.79 | 67.85 |
| | JIT | 3.16 | 7.37 | 3.16 | 7.37 | 2.11 | 8.42 | 5.26 | **23.16** | **8.42** | 22.11 | 1.05 | 8.42 |
| | Combined | 35.18 | 61.32 | 25.20 | 53.23 | 36.39 | 61.46 | 49.73 | 76.82 | **53.50** | **78.71** | 33.96 | 60.24 |
| Combined | Regular | 41.89 | 69.09 | 26.74 | 61.21 | 44.05 | 69.40 | 57.95 | 83.77 | **60.74** | **86.56** | 40.03 | 67.58 |
| | JIT | 36.84 | 63.16 | 24.21 | 56.84 | 33.68 | 68.42 | 47.37 | 81.05 | **48.42** | **86.32** | 35.64 | 61.96 |
| | Combined | 41.24 | 68.33 | 26.42 | 60.65 | 42.72 | 69.27 | 56.60 | 83.42 | **59.16** | **86.52** | 39.65 | 66.73 |

**Action recognition and anticipation with LVLMs**    To benchmark the capabilities of LVLMs for action recognition and anticipation, we reformulated the tasks as image understanding problems. For each video, multiple frames were sampled and composed into a frame sequence image to represent the temporal dynamics visually. The task and action labels were embedded into natural language prompts. We fine-tuned three models, LLaVA-v1.6-7B (Liu et al., 2023; 2024), Qwen2.5-VL-7B (Bai et al., 2025), and Gemma3-4B (Gemma Team et al., 2025), and used the QLoRA (Quantized Low-Rank Adaptation) (Dettmers et al., 2023) approach for efficient adaptation on a single GPU. All models were trained on the regular procedure set and evaluated on regular, JIT, and combined test splits for both recognition and anticipation tasks. As shown in Table 4, LVLMs achieve reasonable performance in action recognition, with LLaVA-v1.6-7B attaining the highest Top-1 accuracy (45.56%) on the Regular test set. Compared to action recognition with VMs, the performances all significantly improve on the JIT set, indicating VLMs' strong zero-shot learning capabilities to unseen and out-of-distribution data. In comparison to VMs like MViT v2 and Uniformer v2 (Table 2 and Table 3), LVLMs lag in anticipation tasks.

## 5.2   BENCHMARK RESULTS FOR VQA

The VQA result can also be seen in Table 4. LLaVa-v1.6(7B) achieved an accuracy of 85.57%, Qwen2.5-VL(7B) with an accuracy of 83.29%, and Gemma3(4B) with an accuracy of 72.04%. Additionally, we tested three smaller VQA models, including ViLT-B/32 (87M), BLIP-base (224M)

Table 4: LVLM Performance (%) on action recognition, anticipation and VQA tasks.

| Task | Action Recognition | | | Action Anticipation | | | VQA |
|------|---------|------|----------|---------|------|----------|------|
| Testset | Regular | JIT | Combined | Regular | JIT | Combined | |
| LLaVa-v1.6-7B | **45.56** | **26.93** | **43.10** | 21.51 | **15.75** | 20.75 | **85.47** |
| Qwen2.5-VL-7B | 42.64 | 24.02 | 40.18 | **23.01** | 13.65 | **21.77** | 83.29 |
| Gemma3-4B | 39.73 | 21.60 | 37.33 | 18.55 | 13.65 | 17.90 | 72.04 |

and Florence-2-base (230M). Among all models, BLIP-base achieves the highest accuracy of 88.64%, demonstrating strong VQA capabilities with a relatively moderate size. Florence-2-base follows closely at 87.86%, while ViLT-B/32 offers a lighter alternative with only 87 million parameters and an accuracy of 79.88%.

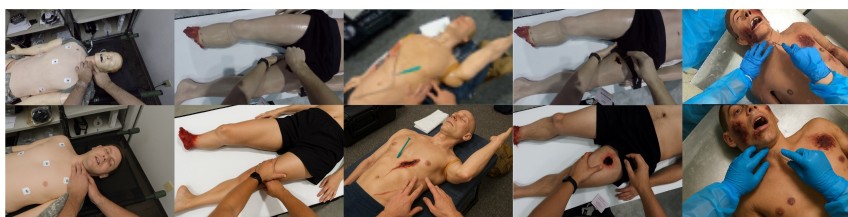

Figure 5: Realistic images generated through our realism framework (TTFW). Top row images are manikin/simulation and the bottom row corresponds to generated images.

## 5.3 REALISM

On our test set, the SVM classifier achieved perfect separability between real-patient and manikin videos, reaching 100% accuracy. To probe its robustness, we incrementally "contaminated" each manikin video by swapping in AI-generated patient frames, measuring the minimum fraction of swapped frames required to flip the SVM's decision. Remarkably, the classifier remained confident in its original (manikin) prediction until roughly two-thirds of the frames had been replaced. On average, 67.34 % of frames needed to be substituted before the model began classifying the video as "real." This result is in accordance with the expectation that at least 50% of frames need to be changed for the classifier to flip its decision. This threshold suggests that the detector relies on a global aggregation of subtle texture and structural cues rather than on a few salient frames, highlighting the high fidelity of our AI-generated images.

In addition to quantitative evaluation, we also performed qualitative comparisons against zero-shot outputs generated without our framework. The generated images frequently failed to maintain semantic fidelity, producing errors such as misplaced hands, altered patient poses, or in some cases failing to generate patients altogether. In contrast, TTFW reliably overcame these shortcomings, delivering medical imagery that was not only realistic but also contextually consistent with the manikin scenes. Figure 5 shows several sample images generated by our framework.

## 6 CONCLUSION

The Trauma THOMPSON dataset provides a comprehensive foundation for advancing AI-driven medical decision-making in austere and resource-constrained environments. By integrating unscripted procedural data, diverse annotations including action recognition, action anticipation, VQA, object detection, hand tracking, and a realism-enhancing framework, it enables the development of unified vision-language models capable of various vision and language tasks. This dataset stands to significantly support the creation of AI copilots that can assist less experienced responders in delivering life-saving care under pressure. The current dataset relies on simplified action annotations (verb + noun). Future efforts will include annotating actions with complete sentences and detailed instructions and collecting videos under more diverse settings, such as variable lighting, adverse weather conditions, and noisy environments.

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

# A  DATASET STATISTICS AND ANALYSIS

## A.1  DATA QUALITY ASSURANCE

To ensure annotation accuracies, the actions in each procedure are annotated by three medical professionals. One person annotates and the other two people review the generated annotations. As the annotations are estimates, and there is no precise way to ensure an absolute timestamp for each procedure, we propose a method to compute the annotation accuracy. Let $t_a$ be the actual timestamp and defined as the average of timestamps from the annotator and the reviewers. $n_r$ is the number of reviewers. $t_{ri}$ is the timestamp from reviewer $i$. $t_o$ is the timestamp from the annotator. $t_a = \frac{1}{n_r+1}(\sum_{i=1}^{n_r} t_{ri} + t_o)$. $t_{as}$ and $t_{ae}$ denote the actual start and end of each clip. $t_{os}$ and $t_{oe}$ denote the original start and end by the annotator. The annotation accuracy of each clip is computed as the overlapping time between the original and actual timestamps divided by the actual clip duration. To compute the overlapping time, we define $t_{start} = \max(t_{os}, t_{as})$ and $t_{end} = \min(t_{oe}, t_{ae})$. The clip accuracy $p_i$ is computed as $\frac{t_{end}-t_{start}}{t_{ae}-t_{as}}$. The average annotation accuracy is computed as $\text{acc} = \frac{\sum_{i=1}^{n}(p_i*(t_{ae}-t_{as}))}{\sum_{i=1}^{n}(t_{ae}-t_{as})}$.

## A.2  ACTION CLASS DISTRIBUTIONS

The dataset comprises 220 videos demonstrating 5 medical procedures and contains 3717 fully annotated video clips. For classification tasks, we have selected class distribution based on the frequency of occurrence in real-world scenarios. The regular procedure includes 42 verb classes, 42 noun classes, and 124 action classes, as illustrated in Figure 6, while the JIT procedure includes 31 verb classes, 37 noun classes, and 101 action classes, as illustrated in Figure 7. The two figures illustrate the imbalanced nature of the dataset. This is because the actions are collected from an unscripted dataset, reflecting the actual frequency of procedures in real life. It is important to note that as the class imbalance in the training data increases, an algorithm's performance tends to decline. An imbalanced dataset makes algorithm development more challenging than a balanced one. Thus, data augmentation techniques will be needed to improve the algorithm's performance, such as oversampling less frequent classes to rebalance the dataset.

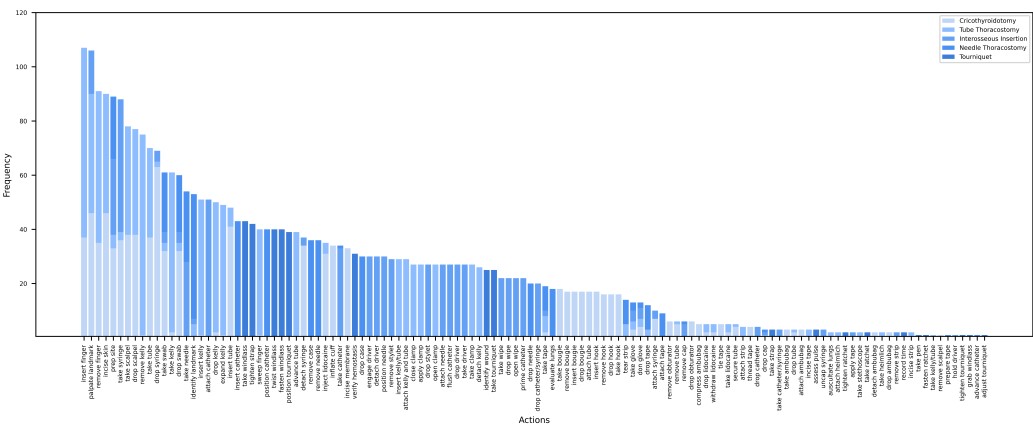

Figure 6: Action classes of regular procedures in the Trauma THOMPSON dataset.

## A.3  ACTION CLASS CO-OCCURRENCES

Figure 8 and 9 depict the frequency of verb-noun combinations within the dataset. Evidently, it can be concluded that the verbs 'take', 'remove', and 'drop' exhibit a higher frequency of co-occurring with nouns. This phenomenon is indeed consistent with the frequency of actions observed during LSI.

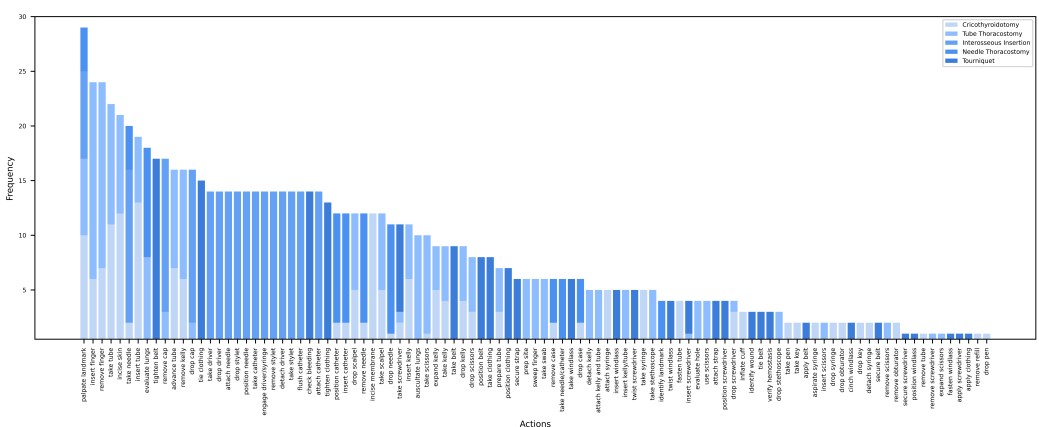

Figure 7: Action classes of JIT procedures in the Trauma THOMPSON dataset.

### A.4 HAND AND OBJECT ANNOTATIONS

To annotate high-quality bounding boxes efficiently, the human-in-the-loop approach is adopted, which combines both manual annotation and automatic tracking. The bounding boxes are created by manual selections of hands and objects in the videos every 10-30 frames and automatically annotated by CSRT trackers Lukežič et al. (2018) between selections. Left hand, right hand, and 15 medical tools are annotated. Teaching VLMs to track hands and recognize objects is clinically significant, especially in high-stakes medical environments. Accurate hand tracking enables AI assistants to assess procedural skills in real-time, offering immediate feedback on bimanual coordination and task execution Azari et al. (2019); Mackenzie et al. (2021). Meanwhile, integrating object detection with natural language understanding allows clinicians to ask AI assistants where specific tools are located, reducing cognitive load and minimizing the risk of human error. Various VLMs have demonstrated object detection capability Feng et al. (2025), such as Florence-2 Xiao et al. (2023) and F-VLM Kuo et al. (2022), highlighting the potential to train unified VLMs that can perform various vision tasks to assist medical procedures.

## B IMPLEMENTATION DETAILS

### B.1 ACTION RECOGNITION AND ACTION ANTICIPATION

The training and testing of action recognition and action anticipation were set up in the same way for all models and only differed on the annotation files. We implemented five models using the open-source MMAction2 library MMAction2 Contributors (2020) (VideoSwin, TimeSFormer, VideoMAE, Uniformer and MViT) and one model (LaViLa) via its GitHub repository. All models were fine-tuned on extracted raw frames to classify 162 classes (regular and JIT actions), initializing from publicly released checkpoints. Input clips were decoded into NCTHW tensors, normalized by the RGB means and standard deviations of the Trauma THOMPSON videos, and then subjected to model-specific augmentations. Unless otherwise noted, we left all other MMAction2 defaults in place.

We fine-tuned the **VideoSwin** model by sampling 32 frame clips at interval 2. During training, each clip was resized and then randomly cropped or horizontally flipped. At inference, we applied a single center crop strategy for validation and a three-crop strategy for testing. We optimized with AdamW using a base learning rate of $10^{-3}$, weight decay of 0.02, warmed up linearly over three epochs, and then cosine-annealed the learning rate across 200 epochs. Training used a batch size of eight and validation used a batch size of four.

We fine-tuned a **TimeSFormer** model by sampling 8 frame clips at interval 32. During training, each clip was randomly rescaled between 256 and 320 pixels, cropped to 224 pixels, and flipped horizontally. At validation, we resized to 256 pixels then center-cropped, while at test we performed three-crop at 224 pixels. Optimization used SGD with a learning rate of 0.005, momentum of 0.9,

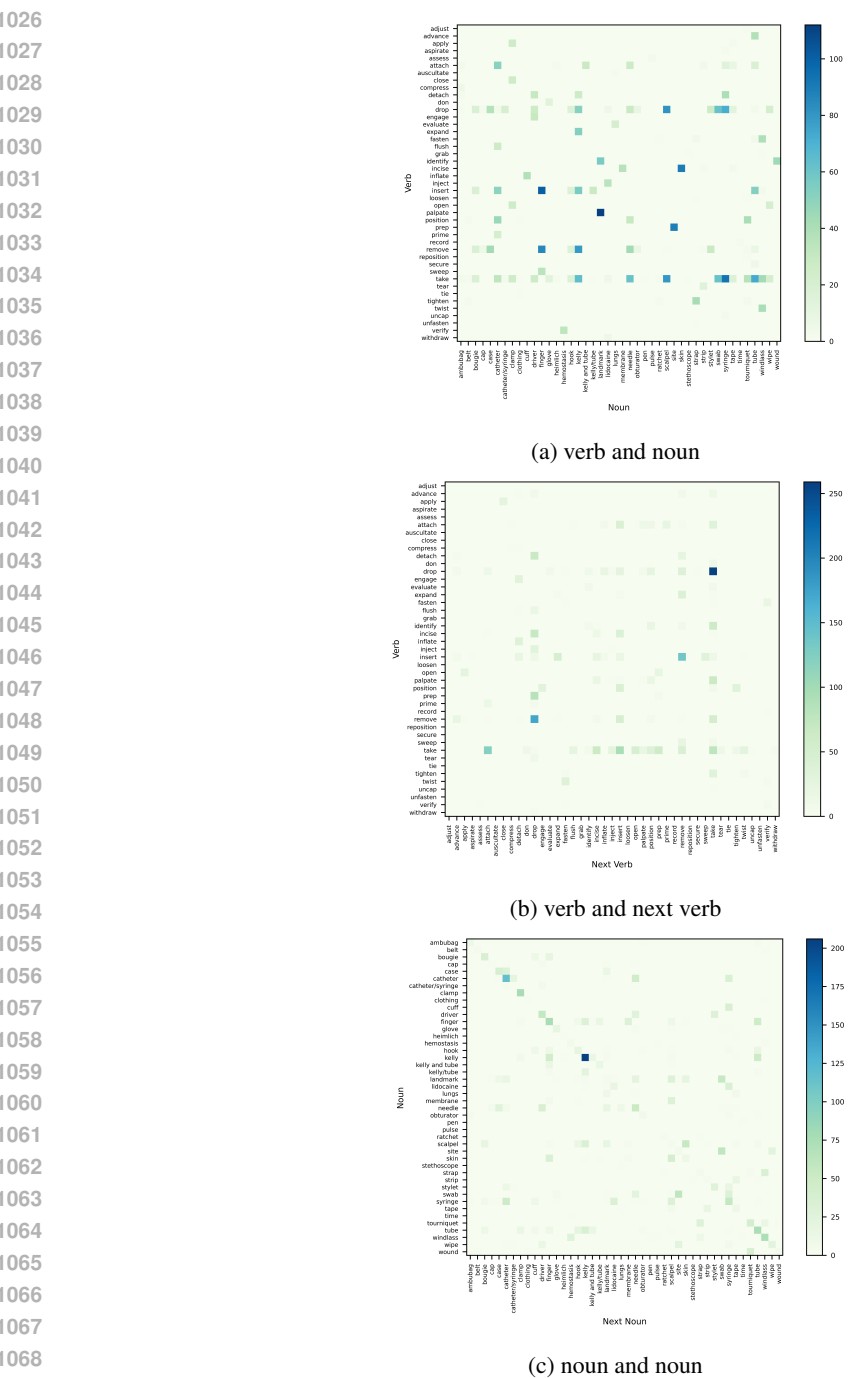

(a) verb and noun

(b) verb and next verb

(c) noun and noun

Figure 8: Frequency of verb noun class co-occurrences of regular procedures.

Nesterov enabled, and weight decay of $10^{-4}$, with no decay on the classification token or positional embeddings. We clipped gradients to a maximum norm of 40 and decayed the learning rate by a factor of ten at epochs 30, 60, 90 and 120 over a total of 200 epochs. Training ran with batch size 12 and validation and testing with batch size 1.

We built the **VideoMAE** model by pairing a 16-frame ViT-Base backbone (16-by-16 patches, 12 layers, 12 heads, 768-dimension embedding) with a TimeSFormer head. Clips were sampled with 16 frames at interval 4 and underwent the same training augmentations as VideoSwin. At validation, we applied a center crop after a 256-pixel resize, and at test, we used five-clip three-crop at 224

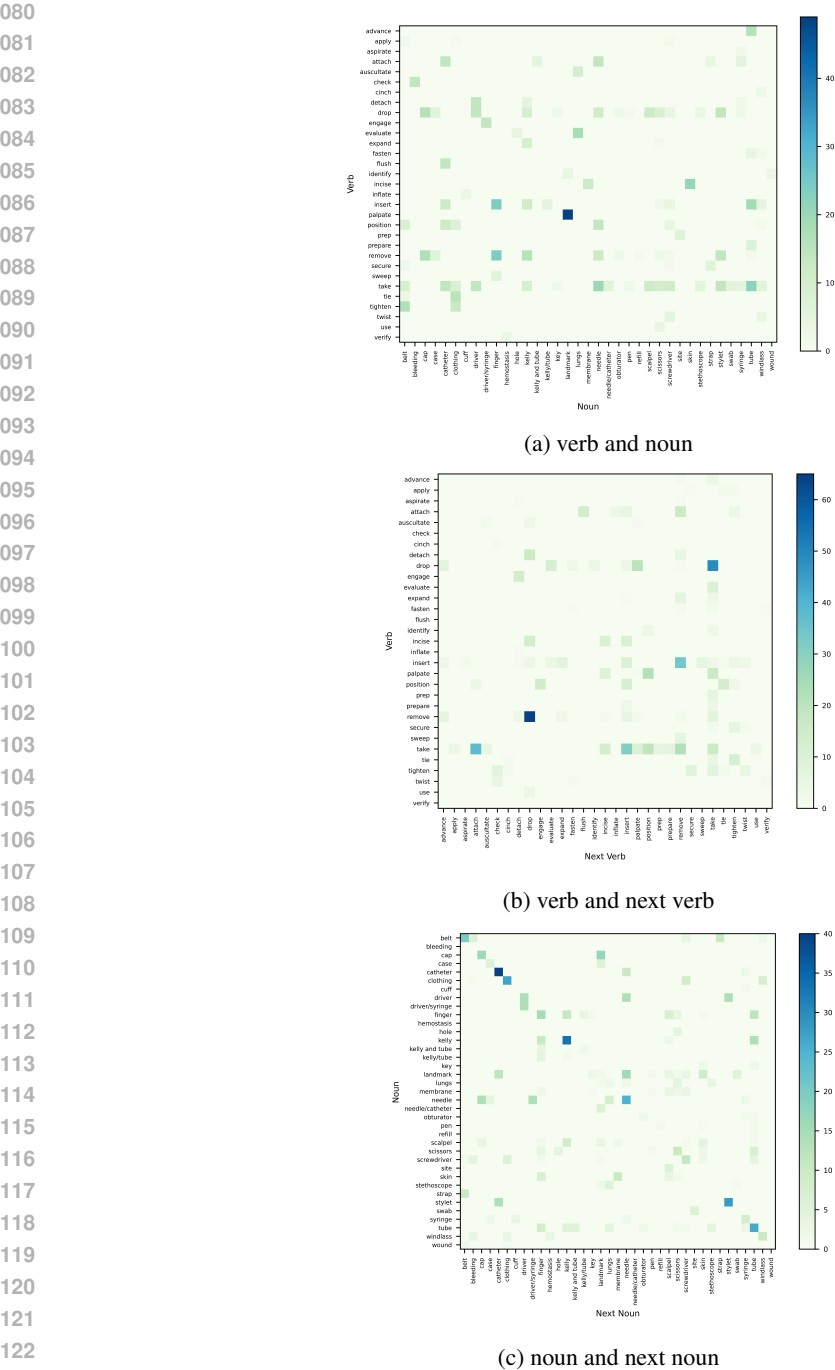

(a) verb and noun

(b) verb and next verb

(c) noun and next noun

Figure 9: Frequency of verb noun class co-occurrences of JIT procedures.

pixels. We optimized with AdamW with learning rate $10^{-3}$, betas 0.9 and 0.999, and weight decay 0.02. All positional embeddings and layer norms were exexempted from decay. A three-epoch linear warmup preceded a cosine-annealing schedule over the remaining 197 epochs, for a total of 200 epochs. Training used batch size four, validation used batch size two, and testing used batch size one.

We fine-tuned a **Uniformer V2** model by initializing its backbone with the following parameters, patch size 16, width 768, 12 layers, 12 heads, and temporal size 32, from Kinetics400 pretrained weights. During training, we sampled 32 frame per clip at interval 2, applied a random resized crop to 224×224 and a 50 percent horizontal flip. At validation, we center-cropped to 224×224, and at test,

we applied three-crop. Inputs were normalized and then fed into a TimeSFormer head with dropout 0.5. We used AdamW optimizer with the same learning rate, betas, weight decay 0.02, positional embeddings, and layer norms as in VideoMAE. Training ran with batch size 6 and validation and testing with batch size 1.

We trained the **MViT V2** model on 16 frames per clip sampled at interval 4. Each clip was resized, augmented via RandAugment with magnitude 7 and 4 layers, then randomly resized and cropped to 224×224, flipped 50 percent, randomly erased 25 percent, and batch blended using Mixup ($\alpha = 0.8$) and CutMix ($\alpha = 1$) before normalization. We wrapped the dataset in a RepeatAugDataset with 2 repeats and optimized by AdamW with a base learning rate $1.6 * 10^{-3}$, betas 0.9 and 0.999, weight decay 0.05. All norms and biases were exempted from decays and gradients were clipped to a max norm of 1. We fine-tuned the model for 200 epochs by warming up the learning rate from 0.016 to $1.6 * 10^{-3}$ over thirty epochs and then cosine-annealing it down to one hundredth of its base value by epoch 200.

We fine-tuned **LaViLa** model with pretrained video–language backbone by inflating its temporal positional embeddings to match our clip length and swapping in a single-head VideoClassifier. Training clips were permuted to channels-first format, randomly resized and cropped to 224×224 with horizontal flips, and normalized per channel. For validation, we used spatial and temporal cropping. Text tokens were processed by LaViLa's default tokenizer. We trained on a single node using AdamW with a learning rate $3 * 10^{-3}$, weight decay 0.05, linear warmup followed by cosine-decay scheduling, gradient accumulation, and clipping. We fine-tuned the model for 100 epochs.

For the training of LVLMs, LLaVa v1.6, Qwen2.5-VL and Gemma3, we sampled 9 evenly spaced frames per video clip and arranged them in a 3x3 grid layout. We used the BLIP model to generate a descriptive caption for each video clip. All models were implemented with the HuggingFace transformer library. The dataset was converted to the chat template specified by the transformer library and augmented by oversampling less frequent classes to balance the training data. We used QLoRA for the three models to allow fine-tuning on a single GPU.

For **LLaVa v1.6**, we configured a 4-bit quantized LLaVa-v1.6-Vicuna-7B model via BitsAnd-BytesConfig and attached a LoRA adapter to all linear modules with rank 16, alpha 16, and 5 percent dropout. An SFTConfig was defined to run 15 epochs of supervised fine-tuning with gradient accumulation, gradient checkpointing, a fused AdamW optimizer under a constant learning-rate schedule, and bfloat16 precision. A custom collate function applied the chat template, tokenized the text, preprocessed the images, padded inputs, and masked label tokens. Finally, we ued TRL's SFTTrainer for the training.

For **Qwen2.5-VL**, we initialized a 4-bit quantized Qwen2.5-VL model via BitsAndBytesConfig and applied a LoRA adapter specifically to the model's query and value projection layers with rank 8, alpha 16, and 5 percent dropout so that most original weights remained frozen. An SFTConfig was set up to run 15 epochs of supervised fine-tuning with gradient accumulation, gradient checkpointing, a fused AdamW optimizer on a constant schedule, and bfloat16 precision. A custom collate function formatted the chat inputs, tokenized text, preprocessed images, and padded sequences and masked both padding and the model's special image-token IDs.

For **Gemma3**, we used a similar setup as LLaVa, a 4-bit quantized Gemma3 model and all-linear LoRA configuration with rank 16 and 5 percent dropout alongside its processor. An SFTConfig was defined to run 15 epochs of supervised fine-tuning with gradient accumulation, a fused AdamW optimizer under a constant learning rate schedule, and bfloat16 precision. The custom collate function pulled out the sequence of messages from each example, applied a BLIP-style processor to tokenize text and preprocess images, and masked padding tokens as well as special image token IDs in the labels.

## B.2 VQA

For the VQA task, we fine-tuned the ViLT model from the GitHub ViLT repository and BLIP, Florence-2, LLaVa v1.6, Qwen2.5-VL, and Gemma3 from HuggingFace's transformer library. For the **ViLT** model, the annotation file was firstly converted from json to arrow format. Before training, the default VQA head was replaced with a lightweight two-layer MLP, projecting the 768-dimensional

backbone output to 1,536 dimensions, then applying LayerNorm and GELU, and finally projecting to the number of answer classes. We fine-tuned the model for 100 epochs.

For the **BLIP** model, a Blip processor tokenized questions and answers with max length padding of 8 and converted images to RGB format. The model was trained for 100 epochs with a batch size 4, learning rate $4 * 10^{-5}$, and mixed precision float16. AdamW was used as the optimizer and an exponential learning rate scheduler was used with gamma set to 0.9.

For the **Florence-2** model, we fine-tuned from florence-2-base-ft checkpoint. We trained it for 50 epochs with a batch size of 1. AdamW was used as the optimizer with a learning rate of $10^{-6}$. A linear learning rate scheduler with no warm up period was implemented.

Additionally, we applied LLaVa v1.6, Qwen2.5-VL, and Gemma3 models on the VQA task. The training and testing configurations were the same as in the action classification tasks, with identical LoRA configurations, optimizer settings, batch size, and precision modes.

## C  MODEL PERFORMANCE BY PROCEDURE TYPES

The radar charts in Figure 10 illustrate the detailed top 1 accuracy of verb, noun, and action of recognizing the five emergency procedures for VideoSwin, TimeSFormer, VideoMAE, Uniformer and MViT, and LaViLa. The similarity of shapes in the radar charts indicates the coherence in the performance of the models for each procedure. It can be seen that no algorithms perform the best in all procedures.

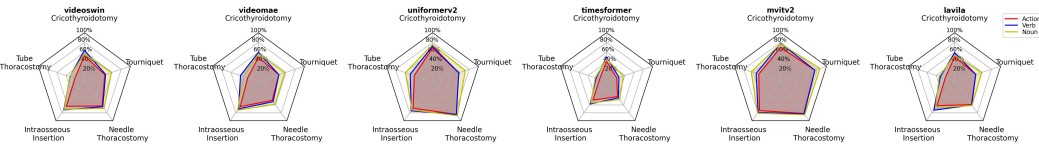

Figure 10: Action recognition Top 1 accuracies of verb, noun, and action by each type of procedure.

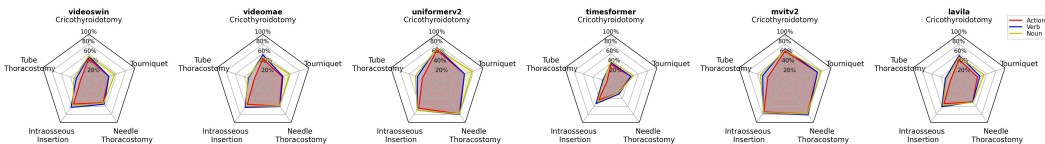

Figure 11: Action anticipation Top 1 accuracies of verb, noun, and action by each type of procedure.

Figure 11 illustrates the comparison for action anticipation on the five emergency procedures. Similar plots are observed in action recognition, but with decreased performance of the models on the action anticipation task.

