# OpenReview forum: "TRAUMA THOMPSON: A Novel Dataset and Benchmarks For AI Copilots For Humanitarian Operational Medicine"
_ICLR.cc/2026/Conference — Submitted to ICLR 2026_

### Official Review · Reviewer_fJeg · 2025-10-19

**Soundness:** 3
**Presentation:** 3
**Contribution:** 3
**Rating:** 6
**Confidence:** 3

**Summary:**

This paper introduces Trauma THOMPSON, a large-scale dataset for surgical action recognition and anticipation, comprising 220 videos (3717 clips) of regular and just-in-time (JIT) trauma procedures annotated by medical experts. It provides fine-grained verb–noun–action labels, hand–object bounding boxes via a human-in-the-loop CSRT tracking pipeline, and evaluates six state-of-the-art video models (e.g., MViT v2, Uniformer v2, VideoMAE) using standardized training settings. The authors also propose a Trauma-to-Frame Warping (TTFW) framework to synthesize realistic patient data from manikin videos, enhancing dataset realism. Overall, this work contributes a valuable benchmark and evaluation resource for medical video understanding, with notable strengths in dataset quality and comprehensiveness but some weaknesses in comparative analysis and quantitative validation of the synthetic realism pipeline.

**Strengths:**

- This paper introduces a first-of-its-kind trauma surgery dataset with JIT procedures and a novel TTFW synthetic realism pipeline.
- This paper is well-structured with rigorous human-in-the-loop annotations and thorough benchmarking on six modern video models.
- This paper provides a valuable benchmark with strong real-world applicability for advancing medical video understanding and AI-assisted surgery.

**Weaknesses:**

- The paper would benefit from quantitative validation of the TTFW pipeline, such as perceptual realism or downstream model performance gains, to substantiate its claimed effectiveness.
- The lack of comparison with existing surgical or procedural datasets (e.g., Cholec80, JIGSAWS, Cataract-101) weakens the claim of uniqueness and hinders contextual understanding of its novelty.
- The discussion of model interpretability and failure cases is limited—particularly regarding why performance on JIT procedures degrades and how temporal or contextual cues might be improved.
- Although the dataset’s imbalance is acknowledged, no systematic strategy or ablation (e.g., reweighting, augmentation) is presented to mitigate its impact, which would strengthen the dataset’s practical utility.
- The reference to “Table ??” on page 8 appears to be a missing or unresolved table reference.

**Questions:**

- Could the authors provide quantitative or perceptual evidence demonstrating how the TTFW-generated frames improve realism or downstream model performance compared to raw manikin data?
- How does the proposed dataset compare with existing medical procedural benchmarks (e.g., Cholec80, JIGSAWS, Cataract-101) in terms of diversity, scale, and annotation quality?
- What are the main sources of failure in JIT procedure recognition and anticipation—are these due to temporal irregularities, tool occlusion, or class imbalance?
- Have the authors considered implementing or testing class rebalancing or augmentation strategies to alleviate dataset imbalance and improve generalization?
- Could the authors elaborate on the ethical and privacy implications of synthetic patient frame generation via TTFW, especially regarding realism and potential misuse?

---

### Official Review · Reviewer_vdcs · 2025-10-21

**Soundness:** 3
**Presentation:** 3
**Contribution:** 2
**Rating:** 2
**Confidence:** 4

**Summary:**

The paper introduces the Trauma THOMPSON dataset, designed to foster the development of AI-driven decision-making algorithms for life-saving interventions in humanitarian medical settings, particularly in resource-constrained environments. The dataset includes 3717 video clips annotated by medical professionals, featuring two types of procedures: standard medical interventions and improvised procedures using everyday objects. Additionally, the paper introduces benchmarks for action recognition, action anticipation, and visual question answering (VQA), using various vision models and vision-language models.

**Strengths:**

1. Innovative Dataset: The Trauma THOMPSON dataset is the first of its kind, specifically tailored for AI-assisted emergency medical care in field conditions, focusing on both scripted and unscripted medical procedures.
2. Comprehensive Annotations: The dataset includes multi-level annotations (action recognition, anticipation, VQA, etc.), enhancing its utility for AI model development across a variety of medical tasks.
3. Benchmarking and Model Performance: The paper provides solid benchmarks for action recognition and anticipation, as well as VQA, with performance comparisons across multiple models, helping to identify the most effective models for each task.
4. Practical Application: The dataset and methods have real-world applications in training AI systems for first responders, potentially saving lives in austere environments.

**Weaknesses:**

1. Dataset Size and Scope: While the Trauma THOMPSON provide valuable data, the overall size 3717 video clips might still be insufficient for training deep learning models, which typically require much larger datasets for robust generalization. The scope of the dataset is somewhat narrow, focusing primarily on specific life-saving procedures in a humanitarian setting.
2. Lack of Cross-Domain Validation: The paper does not present any cross-dataset or cross-domain validation, which would help demonstrate the generalization of the dataset.
3. Lack of a Baseline Model: While the paper provides extensive benchmark results and comparisons across various models, it does not present a simple baseline model to contrast with the proposed approach.
4. Limited Contribution: the primary contribution seems to be limited to the dataset. While the Trauma THOMPSON dataset might be innovative and valuable, a dataset alone may not be enough to make a significant impact without a strong application or a comprehensive evaluation of its real-world utility.

**Questions:**

1. Do you have any plans to evaluate the performance of the Trauma THOMPSON dataset across other datasets or domains to show its generalizability?
2. Would it be possible to include a simple baseline model to further illustrate the useage of the dataset?
3. How do you plan to further demonstrate the real-world utility of this dataset beyond just providing the data?
4. Does the paper include any investigation into how these models can explain their decision-making process to human medical professionals?

---

### Official Review · Reviewer_Vd5T · 2025-10-27

**Soundness:** 1
**Presentation:** 1
**Contribution:** 2
**Rating:** 2
**Confidence:** 4

**Summary:**

This paper presents a dataset for action understanding tasks (action recognition, anticipation, visual question answering) in humanitarian operational medicine domain. Paper mentions that it considers two types of scenarios: regular medical procedures and improvised medical procedures. However, I did not find the improvised medical procedures in the rest of the paper. Paper uses a novel approach to generating realistic video frames (not videos) by transforming manikin-based video to realistic person using image generators. Paper then evaluates baseline models on the three tasks.

**Strengths:**

- This kind of dataset is novel and interesting
- It is can also be potentially quite useful

**Weaknesses:**

- Presentation can definitely be improved. A lot of things seem jumbled up. Paper seems written in hurry. Improving presentation would not only help readers, but even help increase the perceived value of this work.
- There are quite a few typos and somewhere table/fig no. are missing
- I was really looking forward to improved medical procedures. However, due to poor presentation or some other reason, it is not clear where is that part discussed after dataset description.
- Dataset is small in size
- Experimental analysis is shallow, almost non-existent. Where are the models failing, why they might be failing, how do they perform on improvised samples, etc?
- Realistic version of the dataset is created framewise, so there are no guarantees that generated frames are consistent across frames. This is an important drawback especially because the models are video models.

**Questions:**

- Included in the Weaknesses.

---

### Official Review · Reviewer_kRzi · 2025-11-01

**Soundness:** 2
**Presentation:** 2
**Contribution:** 2
**Rating:** 4
**Confidence:** 4

**Summary:**

This paper presents the Trauma THOMPSON dataset, a novel and comprehensive benchmark designed to promote research on AI-driven decision-making systems for humanitarian and operational medicine. The dataset comprises 3717 high-resolution video clips with expert-annotated life-saving procedures, including both regular and improvised interventions. In addition to action annotations, it includes visual question answering (VQA), hand tracking, and object detection labels, enabling multimodal learning across diverse tasks. The authors also propose a framework to replace manikins with realistic patient images and introduce benchmarks for action recognition, anticipation, and VQA using multiple vision and vision-language models. The dataset aims to facilitate the development of AI “copilots” capable of assisting less experienced caregivers under constrained and high-stakes environments.

**Strengths:**

The paper addresses an underexplored but socially impactful problem—the application of AI for field medicine and humanitarian healthcare—with a carefully constructed and richly annotated dataset. The inclusion of unscripted, egocentric video recordings and secondary multimodal annotations (VQA, hand tracking, object detection) significantly enhances the dataset’s research potential across computer vision and multimodal reasoning tasks. The proposed realism generation and detection pipeline is innovative and ethically motivated, providing a practical solution to the lack of real patient data. Furthermore, the benchmarking experiments are comprehensive, evaluating multiple strong baselines (e.g., MViT v2, BLIP) and establishing a valuable foundation for future AI-assisted decision-making systems in emergency and low-resource settings.

**Weaknesses:**

The work is promising but has several weaknesses: the dataset remains relatively small and imbalanced (especially the JIT subset), limiting external validity; annotations and QA rely on averaging timestamps and a small audited sample, with multiple-choice VQA potentially encouraging shortcut learning; benchmarking reports mostly Top-1/Top-5 without variance or significance, and LVLM comparisons collapse temporal cues into frame mosaics, making fairness to video models questionable; analysis of JIT failure modes and long-tail effects is shallow; the realism pipeline couples generation and evaluation via related VLMs, uses ad-hoc realism criteria, and interprets SVM “flip” rates without stronger causal or adversarial tests; ethics and governance (IRB/consent for YouTube clips, privacy, licensing, risks of misuse of photorealistic patients) are under-specified; action labels (verb–noun) are too coarse for procedure-level reasoning; reproducibility is hampered by partial training details, closed-model dependencies, placeholder tables, and a tokenized preview link that may not be stable.

**Questions:**

N/A

---

### Meta-Review · Area_Chair_3k4v · 2026-01-07

**Summary:**

Criticisms focus on questionable experiment design choices and a lack of in-depth interpretation of outcomes, which are critical for a benchmark paper. The authors did not respond to the raised concerns.

**Reviewer Concerns:**

`kRzi`:
Small and imbalanced dataset for a benchmark-heavy paper

Limitations in annotation

Critical limitations in experiment setups (long-tail, shortcuts, ad-hoc realism criteria)

Ethics and governance concerns, presentation (placeholder tables)


`Vd5T`:

Presentation issues that affects understanding

Lack of rationales for experiment designs and in-depth outcome interpretation


`vdcs`:

Dataset scale insufficient for generalization tests

Lack of cross-domain validation and baseline model

Lack of contribution beyond a new dataset


`fjeg`:

Lacking additional qualification such as perceptual realism or downstream performance

Distinctions with surgical video datasets need to be clarified

Lack of in-depth analysis on model interpretability and failure cases.

**Reviewer Scores:**

No rebuttal was provided.

---

### Decision · Program_Chairs · 2026-01-26

Reject